# Validation of the German Version of the Moral Injury Symptom and Support Scale for Health Professionals (G-MISS-HP) and Its Correlation to the Second Victim Phenomenon

**DOI:** 10.3390/ijerph19084857

**Published:** 2022-04-16

**Authors:** Milena Trifunovic-Koenig, Reinhard Strametz, Bianka Gerber, Sneha Mantri, Stefan Bushuven

**Affiliations:** 1Institute for Infection Control and Infection Prevention, Health Care Association District of Constance, 78315 Konstanz, Germany; milenatriko@gmx.de; 2Wiesbaden Institute for Healthcare Economics and Patient Safety, Wiesbaden Business School, RheinMain University of Applied Sciences, 65183 Wiesbaden, Germany; reinhard.strametz@hs-rm.de; 3Institute for Anesthesiology, Intensive Care, Emergency Medicine and Pain Therapy, Hegau Bodensee Hospital Singen, 78224 Singen, Germany; bianka.siebold@gmail.com; 4Department of Neurology and Trent Center for Bioethics, Humanities, and History of Medicine, Duke, University School of Medicine, Durham, NC 27710, USA; sneha.mantri@duke.edu; 5Hegau-Jugendwerk Hospital Gailingen, Health Care Association District of Constance, 78262 Konstanz, Germany; 6Institute for Medical Education, University Hospital, Ludwig Maximilian University of Munich, 80331 Munich, Germany

**Keywords:** moral injury, moral conflict, moral distress, moral stress, moral tension, moral constraint, second victim, medical error, health professionals

## Abstract

Introduction: Comparable to second victim phenomenon (SVP), moral injury (MI) affects health professionals (HP) working in stressful environments. Information on how MI and SVP intercorrelate and their part in a psychological trauma complex is limited. We tested and validated a German version of the Moral Injury Symptom and Support Scale for Health Professionals (G-MISS-HP) instrument, screening for MI and correlated it with the recently developed German version of the Second Victim Experience and Support Tool (G-SVESTR) instrument, testing for SVP. Methods: After translating Moral Injury Symptom and Support Scale for Health Professionals (MISS-HP), we conducted a cross-sectional online survey providing G-MISS-HP and G-SVEST-R to HP. Statistics included Pearson’s interitem correlation, reliability analysis, principal axis factoring and principal components analysis with Promax rotation, confirmatory factor and ROC analyses. Results: A total of 244 persons responded, of whom 156 completed the survey (33% nurses, 16% physicians, 9% geriatric nurses, 7.1% speech and language therapists). Interitem and corrected item-scale correlations did not measure for one item sufficiently. It was, therefore, excluded from further analyses. The nine-item score revealed good reliability (Guttman’s lambda 2 = 0.80; Cronbach’s alpha = 0.79). Factor validity was demonstrated, indicating that a three-factor model from the original study might better represent the data compared with our two-factor model. Positive correlations between G-MISS-HP and G-SVESTR subscales demonstrated convergent validity. ROC revealed sensitivity of 89% and specificity of 63% for G-MISS-HP using a nine-item scale with cutoff value of 28.5 points. Positive and negative predictive values were 62% and 69%, respectively. Subgroup analyses did not reveal any differences. Conclusion: G-MISS-HP with nine items is a valid and reliable testing instrument for moral injury. However, strong intercorrelations of MI and SVP indicate the need for further research on the distinction of these phenomena.

## 1. Introduction

### 1.1. Background

Patient safety includes safe working environments for health professionals (HP) [1]. Post-traumatic stress disorder (PTSD), especially during the COVID-19 pandemic [2], and burnout as an occupational phenomenon [3] are well-known results of stressful, toxic, and unsafe workplaces. This affects specialized post-graduates as well as undergraduates [4]. PTSD alone does not account for all adverse psychological reactions. Moreover, other psychological phenomena could be identified to significantly impact HPs’ well-being. These are moral injury (MI) [5] and the second victim phenomenon (SVP) [6]. Aside from these, other entities, such as “moral stress”, “moral distress”, “moral conflict”, “moral tension”, “moral constraint”, and other terms are used [7,8], and may have comparable effects on HPs, such as PTSD [9].

MI was originally described in the psychological and medical care of military veterans with PTSD [10]. In contrast to PTSD, MI relates to a deep violation of own moral beliefs due to actions which someone has taken or not taken in distinct situations. It comprises emotions and feelings such as shame and the feeling of betrayal by others previously trusted [11]. Compared with moral distress, MI is an acute phenomenon [12]. In contrast, PTSD mainly refers to the experience of life-threatening events, helplessness, extreme anxiety, and existential fear [13]. HPs, such as soldiers, may be exposed to conditions with possible violation of their moral beliefs resulting in harm to physical, psychological, spiritual, and religious integrity, and effects on the well-being of persons themselves and their families. Examples of stressful situations may be found in ethical frontier zones of medicine [14]. This includes end-of-life decisions in critical care [15], triage [16], transplantation [17], abortions [18], decision making in emergency situations, and involvement in economically driven medical practices.

To identify MI among HPs, Mantri et al. developed a screening and detection instrument [19] derived from the original military scale by Koenig et al. [20]. They were able to show that about 23.9% of 181 HPs had MI symptoms and at least moderate functional impairment. This was correlated to young age, low experience, commitments of medical error, symptoms of depression and anxiety, and lack of religious affiliation [21]. By October 2020, less than one year into the pandemic, a global survey of HPs found that nearly half were experiencing functional impairment from MI symptoms [22]. All of these findings raise the demand for further research in this field [23], especially in Germany, with limited literature available focusing on this effect and its relation to the second victim phenomenon [6], with substantial impact on HPs [24] and a prevalence similar to MI [25,26]. Recently, our working group translated SVP instruments [27,28,29,30] into German [31]. Comparing MI and SVP, there are some differences in assessment and definitions. As MI mainly relates to one’s own responsibilities, actions, and decisions, SVP can be acquired by experiencing critical events not caused by the person or an organization the person is affiliated with. However, both show differences from PTSD, which can be accompanied by MI or not [13], and may be caused by SVP [32].

This leads to the question of whether MI and SVP belong to the same psychological entity, or whether they are overlapping but otherwise distinct phenomena. Furthermore, it may be reasonable that both may be part of a multifaceted psychological trauma complex with interacting and overlapping elements. In this paper, we report on the development of the German Moral Injury Symptom Scale (G-MISS-HP) and its correlation to the recently developed German screening score for SVP.

### 1.2. Objective

The aim of this study was to develop and validate a German questionnaire, based on the Moral Injury Symptom Scale for Health Professionals (MISS-HP) [19], to evaluate a cutoff value for clinical use as a screening instrument and to correlate it to the recently developed and validated German version of the Second Victim Experience and Support Tool (G-SVESTR) [31].

We hypothesized that the developed questionnaire shows satisfying reliability, face, content, and construct validity. Furthermore, we hypothesized that there is a strong correlation between scores and their dimensions which is useful for further hypothesis generation to differentiate between MI and SVP.

## 2. Material and Methods

### 2.1. Study Design

From March to June 2021, we designed an online questionnaire consisting of two previously developed questionnaires, namely a translated and back-translated version of the MISS-HP [19] (Appendix A), and G-SVESTR [31]. The study concept followed a cross-sectional approach to validate the translated German version of the MISS-HP and, optionally, correlate answer patterns with the G-SVESTR. 

### 2.2. Setting

Due to the COVID-19 pandemic, with a substantial third wave in Europe in April 2021, the study was conducted online only. Following the WHO advice on the translation of assessment instruments [33], we conducted the expert panels in March to translate, validate, retranslate, and revalidate the instrument. From April to May, we conducted pretesting with 37 individuals. After interviewing, statistical analysis, and re-evaluation in the expert panel, the instrument was distributed unchanged in local, regional, and international media networks and social media networks, addressing about 4500 persons working in in-hospital and out-of-hospital healthcare settings.

### 2.3. Participants

We addressed German-speaking healthcare providers from Germany, Switzerland, Austria, and other countries without any limitation to professions or experience. The only prerequisite to participate was access to the online survey.

### 2.4. Variables and Measurements

Apart from demographic data (age, gender, profession, main occupation, status as a student, status as a medical teacher), the variables comprised the 11 items of the translated MISS-HP, using a 10-point Likert Scale; 42 items of the G-SVESTR, using a 5-Point Likert scale; and an additional free-text entry for comments on the instrument or the topic. Thus, every participant was expected to answer 54 questions. Likert scales of the MISS-HP were the same as in the original scale, ranging from 1 (complete disagreement), through 5 (neutral), to 10 (complete agreement), for items 1–10. Item 11 measured the impairment in everyday work and life caused by the moral injury with a 5-point Likert scaling comprising “none”, “minor”, “moderate”, “severe”, and “extreme”. Participants had to report the extent of agreement/disagreement with all items belonging to G-MISS-HP, i.e., there was no option to omit one of the items and proceed with the questionnaire. Here, we deviated from the original study [19], which allowed for omissions. Likert scales in the SVEST-R part for items 1–35 ranged from 1 (complete disagreement) to 5 (complete agreement). For items 36–42, we used the original 5-point Likert scale for desires of support to SVP from 1 (no strong desire) to 5 (strong desire). Item 43 was a free-text entry.

We were aware that these different scales may have confused some participants. However, we decided to use the original and validated scores according to the primary references of MISS-HP and G-SVESTR.

Concerning the instruments, some variables of both scores were inverted. This applies to items 5, 6, 7, and 10 of the G-MISS-HP, and items 11, 14, 15, 17, 18, 19, 32, 33, 34, and 35 of the G-SVESTR. These items were inverted in their scaling before data processing.

### 2.5. Bias

This study did not aim to display representative occurrence and impact of the MI or SVP in certain closed healthcare worker populations. Instead, we aimed to test the validity and reliability of the scores (without the need for representativeness but demand for enough participants [34]) to plan further research in German-speaking countries.

Furthermore, we used the intercorrelation of the scores to develop new hypotheses concerning the interaction of the two instruments and implications for the distinction of MI and SVP. However, online convenience sampling may bring several barriers to be overcome, mainly related to selection or sample bias: illiteracy or inadequate internet skills may hinder people from participating and sharing surveys in social media may lead to an overrepresentation of specific groups or survey fraud [35].

Further bias includes response burden due to a decline of concentration after a certain number of questions. Pretesting showed a median of 11 min for completion, this is acceptable for online surveys concerning this bias [36].

### 2.6. Study Size

We aimed at a minimum of 10 persons per item completing the survey to validate the G-MISS-HP and conduct confirmatory factor analysis [34]. Of all participants addressed, 244 participants responded to the survey, and 156 fully completed the questionnaire.

### 2.7. Statistics

Prior to performing reliability analyses of G-MISS-HP, we inspected descriptive statics, frequency distributions, and Pearson’s interitem correlation matrix of 10 items (MISS1 to MISS10) of the scale. In addition, we performed reliability analyses using Cronbach’s alpha when each of the items were deleted, which corrected item-scale correlations, and Guttman’s lambda 2. In the next step, we tested the construct validity of the instrument by applying the approach suggested by Boateng et al. [37], to test the factor validity of newly developed instruments. Firstly, we randomly divided the sample into two equal groups. In the first half of the sample, we conducted principal axis factoring (PAF) with Promax rotation, while in the second half, we performed a principal factor analysis (PCA) with Promax rotation. According to the Kaiser–Guttman criterion, we extracted the number of factors with an eigenvalue higher than 1.00. In addition, confirmatory factor analysis (CFA) was used to confirm the factor structure yielded from PAF and PCA. Finally, we compared the differences in model fit of the yielded a two-factor model with the three-factor model proposed by Mantri et al. in the original study [19] using the chi^2^ difference test.

Convergent validity was tested by inspecting Pearson’s correlations between the G-MISS-HP score and the subscales of G-SVSTR. To evaluate the diagnostic properties of G-MISS-HP (cut off values, sensibility, specificity, etc.), we used receiver operating characteristic curve (ROC) analyses. Subgroup analysis was performed using the Mann–Whitney U test to analyze gender differences in the G-MISS-HP score (women vs. men). The Kruskal–Wallis test with Bonferroni correction tested the differences amongst diverse professional groups: nurses (general and pediatric), physicians, paramedics, geriatric nurses, and speech therapists. (The inclusion of other professional groups in the comparison was not feasible due to the small number of participants of other professions.) Finally, Pearson’s correlation coefficients were used to examine the correlation between participant’s age and G-MISS-HP score.

*p*-vales lower than 0.05 were considered statistically significant. Statistical analyses were performed using IBM SPSS Software Version 28 with IBM SPSS AMOS Version 28 (IBM SPSS Statistics for Windows, IBM SPSS AMOS, Version 28.0. Armonk, IBM, New York, NY, USA).

## 3. Results

### 3.1. Sample Characteristics

A total of 156 healthcare professionals were included in the study. Of 156 participants, 100 were women, 54 were men, and 1 was non-binary. There was one missing value stored for this variable in the sample. A total of 147 healthcare professionals worked in Germany, 6 worked in Switzerland, 2 worked in Austria, and 1 worked in New Zealand. The mean age was 40.67 years with *SD* = 10.22 years. One third of all participants were general nursing staff; almost 25% worked as paramedics; 16% were physicians; 9% worked in geriatric nursing; 7.1% were speech and language therapists; 3.2% were medical assistants; 2 participants worked in paediatric care; 2 were medical students; 1 participant worked as a midwife; 1 worked as a respiratory therapist; 1 worked as a physiotherapist; and 1 participant worked in administration. A total of 15 participants were undergoing their training at the time and 60% of the participants were involved in conducting professional training. More than half of all participants worked in hospitals (12.20% in primary healthcare, 19.20% in secondary healthcare, 16% in tertiary healthcare, and 3% in rehabilitation clinics); one quarter worked in pre-hospital emergency medicine/rescue service/patient transport; 8.30% worked in nursing homes; 7.10% worked in palliative care/hospices; 7.10% worked in ambulance; and 1 participant worked in gerontological psychiatry. Table 1 demonstrates sample characteristics alongside descriptive statistics for G-MISS-HP and subscales of G-SVEST-R, with *M(SD*) for continuous and *n* (%) for categorical variables.

### 3.2. Reliability

After reviewing the frequency distributions and descriptive statistics of the G-MISS-HP, we encountered a problem with the inverted item score MISS10 that reads: “*Compared to before I went through these experiences, my religious/spiritual faith has strengthened/Im Vergleich zu der Situation bevor ich diese Erfahrungen gemacht habe, hat sich mein Glaube gefestigt*”. The score of the inverted item MISS10 had the highest mean value (*M* = 7.70) and, unlike other items, had its mode at the upper range of the scale (Mode = 10). The 25% quartile, median and the 75% quartile of the inverted item MISS10 were significantly higher than the corresponding parameters of the other items (see Table 2 and Figure 1).

In addition, we reviewed Pearson’s product–moment correlation matrix of the items. Positive correlations between the items within the scale are necessary to achieve proper internal consistency of the instrument. All items in the G-MISS-HP were positively correlated, except for MISS10, which hardly correlated with any other item on the scale. Furthermore, MISS10 correlated negatively with item MISS9. Negatively correlated items usually suggest that the necessary inversion of the items’ score was not performed, i.e., the numerical scoring scale measured the construct in opposite directions due to differences in positive/negative wording of the items. We verified the original and the inverted values and concluded that the inversion of MISS10 was carried out correctly. Table 3 shows interitem correlations of G-MISS-HP.

Finally, we calculated corrected item-scale correlations altogether with Cronbach’s alpha after excluding the item (see Table 4).

Unlike all the other items, which had corrected item–total correlations above the recommended value of 0.30 [38], MISS10 had a poor negative corrected item–total correlation (−0.07). Therefore, it appeared that, this item should be methodically excluded from the scale. Once we excluded item MISS10, we proceeded with the analysis of item scores for MISS1, MISS2, MISS3, MISS4, MISS8, and MISS9, and inverted item scores MISS5, MISS6, and MSS7.

The internal consistency of the 9 items measured by Cronbach’s alpha was adequate (α = 0.79). Similarly, split-half reliability measured as satisfactory with Guttman’s lambda-2 (λ^2^ = 0.80). In sum, G-MISS-HP instrument with 9 items showed good reliability.

### 3.3. Construct Validity/Factor Analytical Validity

According to the original validation study [19] and the following recommendations for testing factor analytical validity of newly developed scales [39], we randomly divided the sample (*n* = 156) into two halves (*n*_1_ = 78; *n*_2_ = 78). In the first half of the sample, we conducted PAF with Promax rotation, which yielded factor1 (MISS1, MISS2, MISS3, and MISS4) and factor2 (MISS5, MISS6, MISS7, MISS8, and MISS9). These two factors explained 54.7% of the variance (Table 5). In the second half of the sample, we conducted principal factor analysis with Promax rotation, which supported the previously extracted two-factor structure: Factor 1 (MISS1, MISS2, MISS3, and MISS4) and factor 2 (MISS5, MISS6, MISS7, MISS8, and MISS9). Two factors explained 53.32% of the variance (Table 6). In the next step, confirmatory factor analysis (CFA) tested the two-factor model in the whole sample (Figure 2). The parameters for model fit assessment were acceptable (χ^2^_2factors_ = 31.58 *df* = 32, *p* = 0.21, CFI = 0.98, NFI = 0.92, RMSEA = 0.04, IFI = 0.96, AIC = 87.58, ECVI = 0.57). In addition, we tested the three-factor model proposed by Mantri et al., in the validation study of the original English version of the MISS-HP instrument using CFA. The three factors were: (1) the guilt/shame factor (MISS1, MISS2, MISS3, and MISS4), (2) the spiritual troubles factor (MISS5, MISS6, MISS7), and (3) the condemnation factor (MISS8 and MISS9). The three-factor structure showed even better fit indices (χ^2^_3factors_ = 25.31 *df* = 24, *p* = 0.39, CFI = 0.99, NFI = 0.94, RMSEA = 0.02, IFI = 0.99, AIC = 67.31, ECVI = 0.43, see Figure 3). To determinate whether the two-factor or the three-factor model was a better representation of the data, we performed the chi^2^ difference test:∆χ^2^ = χ^2^_2factors_ − χ^2^_3factors_ = 31.58 − 25.31 = 6.27; ∆χ^2^(df) = *df*_2Factors_ − *df*_3Factors_ = 26 − 24 = 2.
where ∆χ^2^_diff_(2) = 6.27 was significant (*p* = 0.04).

Therefore, the three-factor model, from the original study [19], was more appropriate, evidencing construct validity (see Table 7).

### 3.4. Convergent Validity

Convergent validity was demonstrated by significant correlations between G-MISS-HP sum score (9 items, see above) and G-SVESTR subscales (see Table 8), whereas strong correlations existed between G-MISS-HP score and following G-SVESTR subscales: psychological distress (*r* = 0.55), physical distress (*r* = 0.59), professional self-efficacy (*r* = 0.56), and turnover intentions (*r* = 0.59). G-MISS-HP was moderately correlated to colleague support (*r* = 0.49), institutional support (*r* = 0.33), absenteeism (*r*= 0.44), and resilience (*r* = 0.33). The only weak correlation was between G-MISS-HP score and supervisor support (*r* = 0.29).

### 3.5. Diagnostic Properties of G-MISS-HP

ROC analyses were performed using G-MISS-HP score (with 9 items) as a test variable, with the following dichotomized item as a state variable: “Do the feelings you indicated above cause you significant distress or impair your ability to function in relationships, at work, or other areas of life important to you? In other words, if you indicated any problems above, how difficult have these problems made it for you to do your work, take care of things at home, or get along with other people?” The response options “moderate”, “very much”, or “extremely” indicated a positive actual state, i.e., significant impairment in everyday functioning caused by MI. In contrast, response options “not at all” or “mild” indicated negative actual state [40]. Figure 4 shows the ROC curve. The area under the curve was 0.81 (asymptotic 95% CI = 0.75–0.88), standard error under the non-parametric assumption was 0.03, and the asymptotic significance was *p*≤ 0.001 (under the null hypothesis that true area = 0.5). After inspecting the Youden index for different coordinates of the ROC curve, we determined the optimal cutoff score on the G-MISS-HP as 28.50. This means that healthcare professionals with a summative G-MISS-HP score of 28.50 or higher were tested positive, indicating significant impairment caused by MI. The sensitivity of G-MISS-HP with the cutoff 28.5 was good with 89%. In contrast, the specificity of the instrument was low, with 63%, but still acceptable for a potential screening instrument. The positive predictive value (PPV) was 62% (of the 85 with a positive test, 53 were impaired), and the negative predictive value was 69% (of the 103 with a negative test, 71 were without impairment).

### 3.6. Subgroup Analyses

There was no significant correlation between participants’ age and G-MISS-HP sum score (*r* = −0.009; *p* = 0.92). Gender difference in G-MISS-HP score was also not significant (Mann–Whitney U test: *z* (2, 154) = −1.75; *p* = 0.09). Moreover, the G-MISS-HP score did not differ significantly across different professions (Kruskal–Wallis test with Bonferroni correction *H* (5) = 10.01; *p* = 0.08).

## 4. Discussion

We demonstrated that the German Version of the MISS-HP is a reliable and valid instrument, comprising face, content, and construct validity. Internal consistency, as well as split-half reliability analyses, yielded satisfactory results. Unfortunately, item MISS10 was problematic with a need for reassessment or removal from the sum score.

Item MISS10 addresses the beliefs in a God. Recent findings confirmed a negative-trend-level association between religiosity and MISS-HP score [22]. Therefore, we presumed that the possibly higher proportion of atheism among German healthcare workers, compared with the USA, might explain the poor item-scale statistics of MISS10. Contrary to our assumptions, the proportion of agnostics or atheists amongst healthcare providers in Germany and in the USA do not differ considerably. In Germany, up to 25% of all healthcare providers fall into the category of atheists/agnostics or do not feel attached to religious groups [41], while in the USA, 24% of physicians proclaimed themselves as agnostic or atheist [41].

Thus, item MISS10 could not be answered validly by atheists as there is no belief to be strengthened or weakened. Since there was no option to not respond to an item and proceed with the questionnaire afterward, it is questionable how atheistic participants answered our obligatory question (1—fully disagree; 5—neutral points). This might have provoked a significant bias. However, in the validation study of the English version by Mantri et al., item MISS10 could be omitted by respondents, and analyses were conducted by extrapolating missing data [19].

Courtesy of the authors, we obtained the raw data from the study by Mantri et al. [19]. In addition, we inspected the frequency distribution of MISS10 in the dataset (see Figure 5). Thereby, we noticed a considerably large number of missing values regarding this variable. The dataset comprised two different types of item non-responders: “the participant did not even see the question (e.g., had started the survey but did not hit the “next page” button)” and “the participant saw the question but did not answer”. We assumed that the considerable proportion of the participants in the latter group might not have responded to the item due to a lack of personal religious affiliation. Moreover, if the missing values of “saw the question but did not answer” were assigned a value of 1 (fully disagree), then the frequency distribution of the item score would be very similar to the frequency distribution in the present study (see Figure 6 and Figure 7). Therefore, different ways to deal with potential item non-responders in English and German versions had possibly contributed to the inconsistencies regarding the suitability of MISS10 for the instrument.

To get further insight into the validity of the 10-item score, we excluded all participants that chose 1 point (fully disagree) in item MISS9 as a surrogate for atheism. In addition, we re-evaluated the 10-item score. In this analysis, the 10-item score showed good overall properties (Cronbach’s Alpha 0.79; Gutman’s lambda 0.81; weak positive corrected interitem correlation 0.04) and demonstrated a 3-factor solution after performing PCA. However, choosing a value of 1 in item MISS9 does not necessarily indicate atheism, lowering the validity of the approach taken.

Although these findings may be considered a setback for the direct translation of the score, it raises the important question of how religious beliefs and practices should be addressed in MI and SVP scores. Thus, an alternative method could rather entail the following: (1) life conceptions, identity issues, or religious beliefs (“internal”); and (2) socialization and integration in social groups (“external”)—these interacting factors have a significant impact on coping [42].

In sum, item MISS10, which assesses religious beliefs and participation in religious practice, should be re-evaluated. Nevertheless, the reduced 9-item score can be used validly. However, it may lack content comparability to the English survey with the 10-item score. In multinational investigations with the need to compare the scores in different languages, item MISS10 could remain but should be provided as an optional question, e.g., the item should be presented only to participants who previously reported themselves as religious/spiritual.

Secondly, the observed test statistic of the 9-item sum score consequently showed lower cutoff values indicating MI (28.5 points with 89% sensitivity, but only 64% specificity) compared with the English 10-item MISS-HP Score (36 points with 84% sensitivity and 94% specificity) [19]. A relatively high sensitivity provides a strong argument that G-MISS-HP can be used as a screening instrument. Nevertheless, poor specificity of the instrument potentially leads to high false-positive rates. Hence, the instrument’s capacity to correctly rule out clinically relevant impairment caused by MI is significantly limited. However, as the original English MISS-HP was mainly validated by physicians, and the German G-MISS-HP in a multi-professional setting, further work should concentrate on inter-professional differences in bigger sample sizes [34]. Possibly, the specificity of the score might differ in distinct professions or professional groups.

Thirdly, the G-MISS-HP showed good comparability to the German Second Victim Experience and Support Scale Revised (G-SVESTR). This indicates that MI and SVP are related constructs and that both scores may be used as screening instruments after further evaluations. Nevertheless, none of the correlations between the G-MISS-HP Score and G-SVESTR subscales was higher than *r* = 0.60. In conclusion, SVP and MI seem to be distinct but interacting phenomena. Further research is needed to explain the exact relationship between the constructs.

However, subgroup analysis showed no differences for gender, age, or work experience. This is in contrast to the SeViD questionnaires for physicians [26] and nurses [43] and recent findings of epidemiological studies exploring the prevalence of MI in the US [21]. Those studies showed differences in these subgroups and a significant negative correlation between age and the prevalence of MI.

Furthermore, this study is not able to reveal sociodemographic or epidemiological data on MI or SVP in the distinct populations addressed. Primarily, spiritual, religious, and cultural assessments should be conducted in future investigations. These may reveal further interpretation possibilities of the score in different groups and professions. The aim of this study was to validate the score with an adequate sample size [34] for future analysis in different populations. In addition to the lack of generalizability so far, further limitations should be acknowledged. These include the sample size [34] and the importance of the convenience of the online approach for validation [35].

Nevertheless, MISS-HP may be a promising instrument for detecting MI in changing and developing medical environments and settings [44]. As medical knowledge and technology advance and population demographics change rapidly, new ethical questions will arise and challenge medical professionals from different countries and cultures [45,46,47]. This may be the case at the frontiers of life (e.g., neonatology, neuro-traumatology, geriatrics, and genetics). Consequently, medical professionals may experience more moral conflicts and more errors, putting staff at risk for both MI and SVP with need for an assessment tool for supervisors, crew resource managers, and psychotherapists treating traumatized staff. G-MISS-HP may be one of these tools.

## 5. Conclusions

The nine-item German Moral Injury Assessment and Support Score for Health Professionals (G-MISS-HP) is a reliable and valid instrument to screen for MI. We could show intercorrelations with SVP, indicating a relationship between these two phenomena. This first evaluation did not include epidemiologic or demographic statements. Further work should concentrate on the deeper distinction of MI and SVP, the epidemiology of MI and SVP in Germany, and the status of spiritual and religious aspects within MI and SVP scoring systems.

## Figures and Tables

**Figure 1 ijerph-19-04857-f001:**
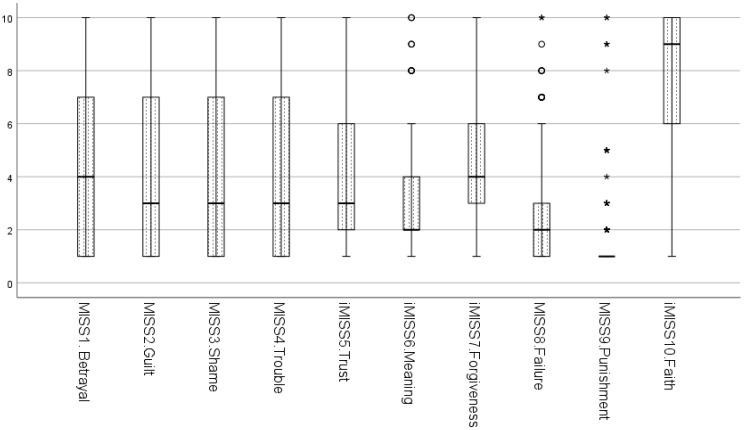
Boxplots of ten items of G-MISS-HP. Note. G-MISS-HP—Moral Injury Symptom Scale for Health Professionals, German version; MISS5i, MISS6i, MISS7i, MISS10i—inverted items scores due to positive wording.

**Figure 2 ijerph-19-04857-f002:**
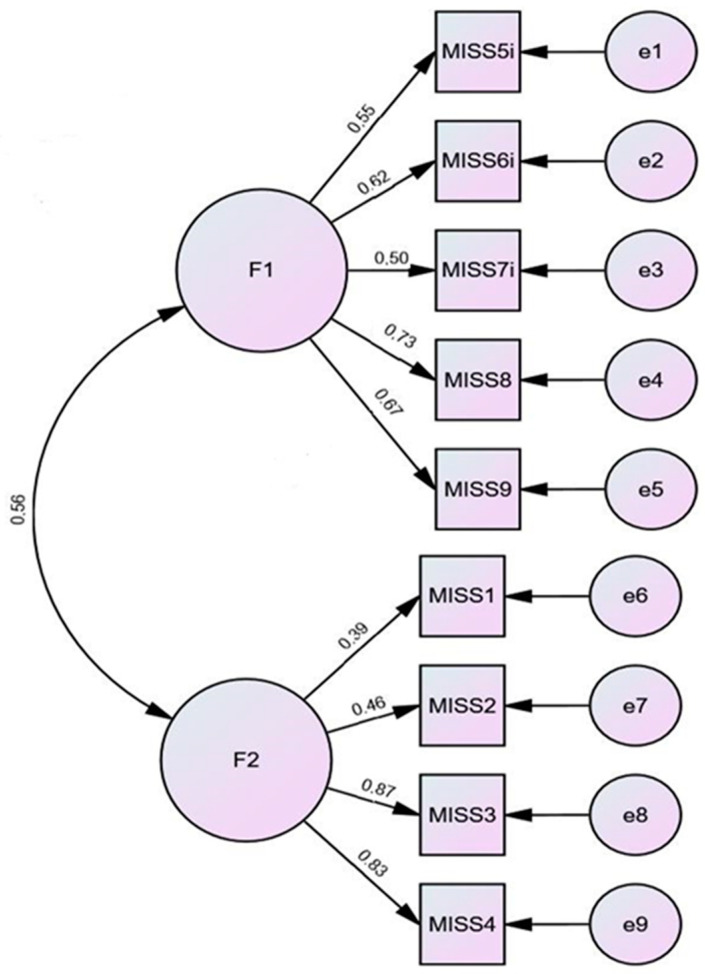
Confirmatory factor analysis of G-MISS-HP; 2-factor model. Note. Values on paths correspond to standardized estimates.

**Figure 3 ijerph-19-04857-f003:**
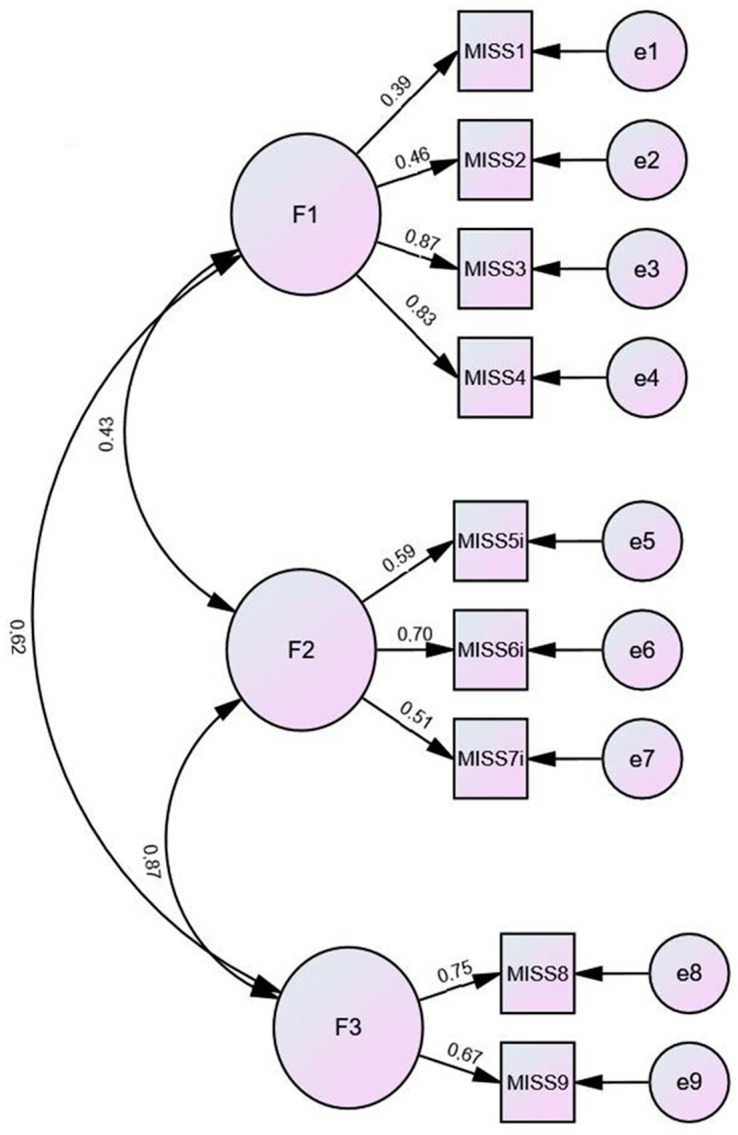
Confirmatory factor analysis of G-MISS-HP; 3-factor model. Note. Values on paths correspond to standardized estimates.

**Figure 4 ijerph-19-04857-f004:**
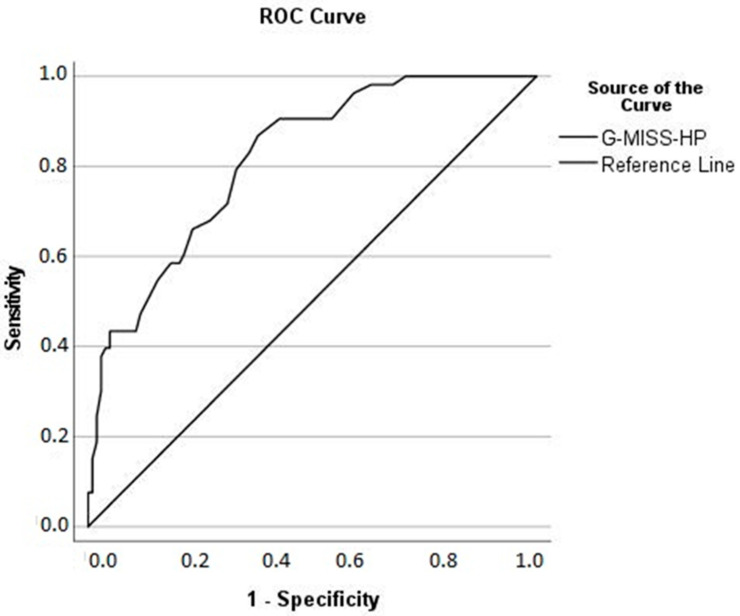
Receiver operator curve (ROC) for G-MISS-HP.

**Figure 5 ijerph-19-04857-f005:**
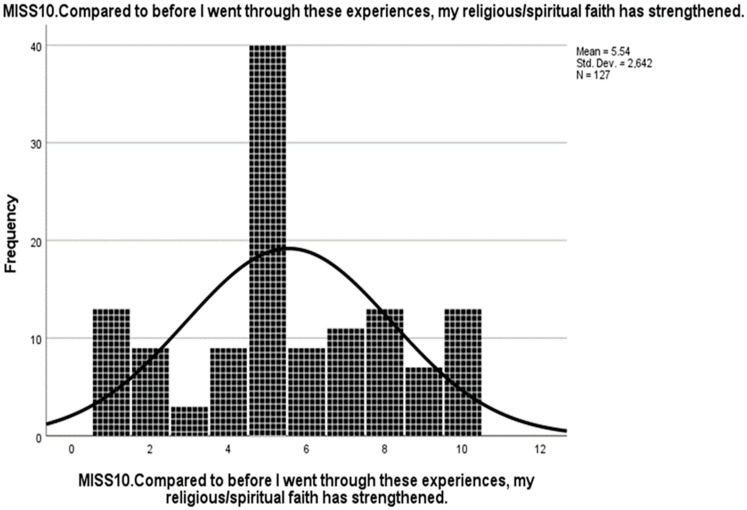
Frequency distribution of MISS10 in the original study by Mantri et al., when the participants, who omitted the item after they had read it, were treated as missing values [19].

**Figure 6 ijerph-19-04857-f006:**
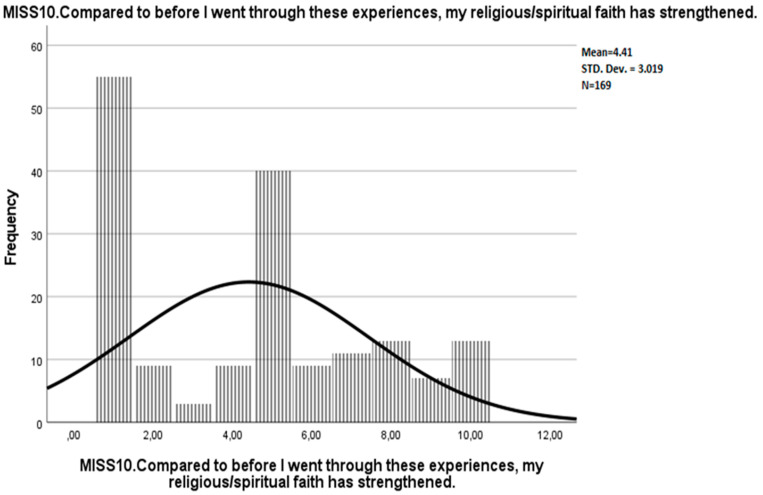
Frequency distribution of MISS10 in the original study by Mantri et al., when the participants who omitted the item after they had read it, were assigned a value of 1 [19].

**Figure 7 ijerph-19-04857-f007:**
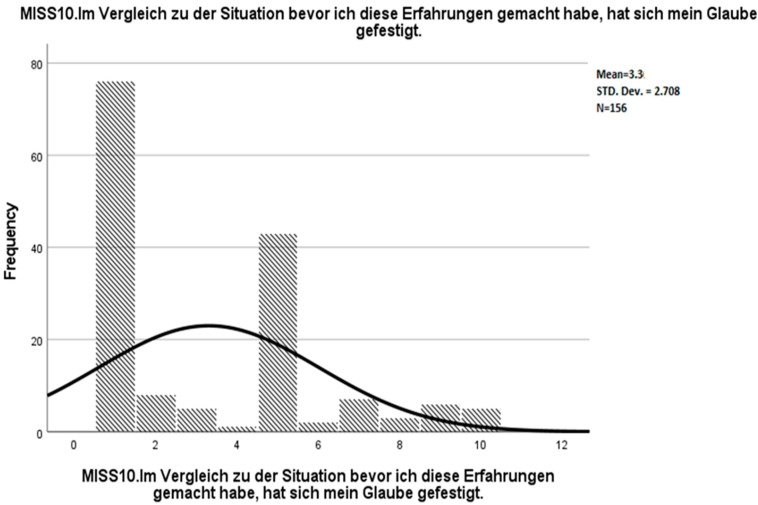
Frequency distribution of MISS10 in the present study, German version.

**Table 1 ijerph-19-04857-t001:** Sample characteristics and descriptive statistics of the G-MISS-HP variables and G-SVESTR subscales.

Characteristics		% (*n*)	*M* (*SD*)
Sociodemographic			
Age(years)			40.67 (10.22)
Gender	Female	64.1% (100)	
	Male	34.6% (54)	
	Non-binary	0.6% (1)	
National Workplace	Germany	94.2% (147)	
	Austria	1.3% (2)	
	Switzerland	3.8% (6)	
	New Zeeland	0.6% (1)	
Profession	Physician	16% (25)	
	Nurse (general)	33.3% (52)	
	Paediatric nurse	1.3% (2)	
	Paramedic	23.7% (37)	
	Geriatric nurse	9% (14)	
	Speech therapist (clinical linguistic)	7.1% (11)	
	Medical assistant	3.2% (5)	
	Medical student	1.3% (2)	
	Physiotherapist	0.6% (1)	
	Midwife	0.6% (1)	
	Respiratory therapist	1.3% (2)	
	Administration worker/Technician	0.6% (1)	
Working area	Primary hospital	12.2% (19)	
	Secondary hospital	16.6% (26)	
	Tertiary hospital	19.2% (30)	
	Rehabilitation clinic	3.2% (5)	
	Pre-hospital emergency medicine/rescue service/patient transport	25.6% (40)	
	Nursing homes	8.3% (13)	
	Palliative care/hospices	7.1% (11)	
	Ambulance	7.1% (11)	
	Gerontological psychiatry	0.6% (1)	
Professional training	Completed	89.7% (140)	
	Trainee	9.6% (15)	
	Trainer	60.3% (94)	
G-MISS-HP	9 Items (MISS1-MISS9)		32.31 (13.26)
G-SVESTR	Psychological distress		2.92 (1.04)
	Physical distress		2.92 (1.04)
	Colleague support		2.54 (1.10)
	Supervisor support		1.98 (0.65)
	Institutional support		2.68 (1.21)
	Professional self-efficacy		3.25 (1.18)
	Turnover intentions		2.25 (1.07)
	Absenteeism		2.20 (1.24)
	Resilience		1.90 (1.02)

Note. G-MISS-HP—Moral Injury Symptom Scale for Health Professionals, German version; G-SVESTR—Second Victim Experience and Support tool, German version.

**Table 2 ijerph-19-04857-t002:** Descriptive item statistics of G-MISS-HP.

Item	Mean	Mode	Standard Deviance
MISS1—Betrayal	4.04	1.00	2.57
MISS2—Guilt	3.86	1.00	2.67
MISS3—Shame	4.31	1.00	2.93
MISS4—Trouble	4.26	1.00	2.98
MISS5—Trust ^1^	3.97	2.00	2.34
MISS6—Meaning ^1^	3.03	2.00	1.91
MISS7—Forgiveness ^1^	4.62	6.00	2.34
MISS8—Failure	2.53	1.00	2.07
MISS9—Punishment	1.68	1.00	1.78
MISS10—Faith ^1^	7.70	10.00	2.71

Note. G-MISS-HP—Moral Injury Symptom Scale for Health Professionals, German version; ^1^—inverted item scores due to positive wording.

**Table 3 ijerph-19-04857-t003:** G-MISS-HP interitem correlation matrix (Pearson’s correlation coefficients).

Item	MISS1	MISS2	MISS3	MISS4	MISS5 ^1^	MISS6 ^1^	MISS7 ^1^	MISS8	MISS9	MISS10 ^1^
MISS1—Betrayal	1									
MISS2—Guilt	0.16 **	1								
MISS3—Shame	0.32 **	0.41 **	1							
MISS4—Trouble	0.35 **	0.35 **	0.73 **	1						
MISS5—Trust ^1^	0.26 **	0.10	0.22 **	0.25 **	1					
MISS6—Meaning ^1^	0.08	0.15	0.26 **	0.19 *	0.45 **	1				
MISS7—Forgiveness ^1^	0.14	0.17 *	0.20 *	0.19 *	0.25 **	0.34 **	1			
MISS8—Failure	0.20 *	0.35 **	0.40	0.35 **	0.34 **	0.47 **	0.36 **	1		
MISS9—Punishment	0.14	0.17 *	0.36 **	0.35 **	0.39 **	0.37 **	0.35 **	0.50 **	1	
MISS10—Faith ^1^	−0.02	−0.09	−0.05	0.001	−0.01	0.09	−0.01	−0.11	−0.23 **	1

Note. ^1^—inverted item score due to positive wording; *—the correlation is significant at the *p* < 0.05 level (two-tailed); **—the correlation is significant at the *p* < 0.01 level (two-tailed).

**Table 4 ijerph-19-04857-t004:** Reliability analysis of G-MISS-HP.

Item	Scale Mean If Item Deleted	Scale Variance If Item Deleted	Corrected Item–Total Correlation	Cronbach’s Alpha If Item Deleted
MISS1—Betrayal	35.96	150.24	0.34	0.72
MISS2—Guilt	36.15	147.62	0.36	0.72
MISS3—Shame	35.70	129.28	0.60	0.67
MISS4—Trouble	35.74	129.08	0.59	0.68
MISS5—Trust ^1^	36.03	148.61	0.42	0.71
MISS6—Meaning ^1^	36.97	152.90	0.46	0.71
MISS7—Forgiveness ^1^	35.39	151.27	0.37	0.72
MISS8—Failure	37.47	145.71	0.56	0.69
MISS9—Punishment	38.33	154.44	0.46	0.71
MISS10—Faith ^1^	32.31	175.85	−0.07	0.79

Note. G-MISS-HP—Moral Injury Symptom Scale for Health Professionals, German version; ^1^—inverted item score due to positive wording.

**Table 5 ijerph-19-04857-t005:** Structural matrix factor loadings of principal axis factoring analysis with Promax rotation for 9 items of G-MISS-HP (*n*_1_ = 78; first half of the sample).

Item	Factor 1	Factor 2
MISS1—Betrayal	0.10	**0.40**
MISS2—Guilt	0.06	**0.43**
MISS3—Shame	0.32	**0.85**
MISS4—Trouble	0.31	**0.93**
MISS5—Trust ^1^	**0.61**	0.08
MISS6—Meaning ^1^	**0.78**	0.15
MISS7—Forgiveness ^1^	**0.47**	0.09
MISS8—Failure	**0.68**	0.46
MISS9—Punishment	**0.79**	0.41
Initial eigenvalues	3.44	1.33
Kaiser–Meyer–Olkin (KMO) = 0.79		
Bartlett’s Test of Sphericity = χ^2^ = 384.24; *p* < 0.001		

Note. G-MISS-HP—Moral Injury Symptom Scale for Health Professionals, German version; ^1^—inverted item scores due to positive wording. Bold factor loadings indicate the factor assigned.

**Table 6 ijerph-19-04857-t006:** Structural matrix factor loadings of principal components analysis with promax rotation for 9 items of G-MISS-HP (n_2_ = 78; second half of the sample).

Item	Factor 1	Factor 2
MISS1—Betrayal	*0.21*	** *0.55* **
MISS2—Guilt	*0.25*	** *0.63* **
MISS3—Shame	*0.40*	** *0.87* **
MISS4—Trouble	*0.37*	** *0.86* **
MISS5—Trust ^1^	** *0.74* **	*0.51*
MISS6—Meaning ^1^	** *0.72* **	*0.41*
MISS7—Forgiveness ^1^	** *0.69* **	*0.27*
MISS8—Failure	** *0.77* **	*0.21*
MISS9—Punishment	** *0.63* **	*0.24*
Initial eigenvalues	3.70	0.90
*Kaiser–Meyer–Olkin (KMO) = 0.81*		
*Bartlett’s Test of Sphericity = χ^2^ = 188.57; p < 0.001*		

Note. G-MISS-HP—Moral Injury Symptom Scale for Health Professionals, German version; ^1^—inverted item scores due to positive wording. Bold factor loadings indicate the factor assigned.

**Table 7 ijerph-19-04857-t007:** Fit indices and chi-square difference tests for two different models (two-factor models vs. three-factor models).

Model	χ^2^(df)	∆χ^2^(df)	*p*	CFI	NFI	RMSEA	IFI	AIC	ECVI
2 Factors	31.58 (32)		0.21	0.98	0.92	0.04	0.96	87.56	0.57
3 Factors	25.31 (32)		0.39	0.99	0.94	0.02	0.99	67.31	0.43
2 Factors vs. 3 Factors		6.72 (2)	0.04						

Note. CFI—comparative fit index; NFI—normed fit index; RMSEA—root mean square error of approximation; IFI—incremental fit index; AIC—Akaike information criterion; ECVI—expected cross validation index.

**Table 8 ijerph-19-04857-t008:** Pearson’s correlations between G-MISS-HP Score and G-SVESTR subscales.

Variable	1	2	3	4	5	6	7	8	9	10
1. G-MISS-HP	--									
2. Psychological distress	0.55 **	--								
3. Physical distress	0.59 **	0.65 **	--							
4. Colleague support	0.49 **	0.35 **	0.37 **	--						
5. Supervisor support	0.29 **	0.17 *	0.28 **	0.48 **	--					
6. Institutional support	0.33 **	0.36 **	0.28 **	0.47 **	0.40 **	--				
7. Professional self-efficacy	0.58 **	0.65 **	0.58 **	0.45 **	0.29 **	0.38 **	--			
8. Turnover intentions	0.59 **	0.48 **	0.62 **	0.53 **	0.51 **	0.49 *	0.56 **	--		
9. Absenteeism	0.44 **	0.43 **	0.59 **	0.27 *	0.33 **	0.20 *	0.45 **	0.54 **	--	
10. Resilience	0.33 **	0.05	0.12	0.19 *	0.24 *	0.13	0.22 **	0.25 **	0.19 *	--

Note. G-MISS-HP—Moral Injury Symptom Scale for Health Professionals, German version; **—correlation is significant at the 0.01 level (2-tailed); *—correlation is significant at the 0.05 level (2-tailed).

## Data Availability

The data presented in this study are available on request from the corresponding author.

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
