# Peer review of "Validation of the German Version of the Moral Injury Symptom and Support Scale for Health Professionals (G-MISS-HP) and Its Correlation to the Second Victim Phenomenon"

_ijerph, 2022, doi:10.3390/ijerph19084857_

Round 1
Reviewer 1 Report
Thank you for letting me review this interesting paper, I ve really appreciated the presented methodology and I think that it could be of a great interest for an international reader, I just suggest the authors to read and include in the references the following papers:
Medico-legal assessment of personal damage in older people: report from a multidisciplinary consensus
conference
International Journal of Legal Medicine. 2020 Nov;134(6):2319-2334.doi: 10.1007/s00414-020-02368-z.
Author Response
Dear reviewer
many thanks for your advice. We included the citation and some others in the discussion section.
Reviewer 2 Report
Dear authors,
after multiple readings of your manuscript, I find the study well conducted and very well described. This works surely will add value to the second victim literature and open new research paths towards a better understanding of the phenomenon.
Kind Regards,
Author Response
We thank your for your comments!
Reviewer 3 Report
This is a validation study. In my opinion the topic is pertinent and these instruments are useful. The methods applied are right. There are some points that authors could consider:
- Limitation section must be included explaining main aspects to be considered to a better understanding of the results.
- A back-translation procedure has been applied. This method is right, but more details about cross-cultural validity could be included.
- Some details about the interpretability of the scores could also be useful.
- Discussion is right and relevant information is included. However, this section could also included a reference about how, when and how the instrument could be used. What users can expect using this instrument should be explained.
Author Response
Dear reviewer
Many thanks for your advice!
First, we included a limitation section in the discussion
Second, we included this point in the discussion section concerning demands for future investigations.
Third, we mentioned the need for more demographic data to allow more interpretations. This was done in the discussion section as well. By now, the interpretability of the score is limited and was not the primary aim of the validation process. We will concentrate on this in future studies with the score.
Fourth, we included a section in the discussion how an where the score can be potentially used.